# New Drugs Bringing New Challenges to AML: A Brief Review

**DOI:** 10.3390/jpm11101003

**Published:** 2021-10-03

**Authors:** Zhi Han Yeoh, Ashish Bajel, Andrew H. Wei

**Affiliations:** 1Department of Clinical Haematology, Peter MacCallum Cancer Centre & The Royal Melbourne Hospital, Melbourne, VIC 3000, Australia; zhihan.yeoh@petermac.org (Z.H.Y.); ashish.bajel@mh.org.au (A.B.); 2The Sir Peter MacCallum Department of Oncology, University of Melbourne, Melbourne, VIC 3000, Australia; 3Department of Clinical Haematology, Alfred Hospital and Monash University, Melbourne, VIC 3004, Australia; 4Division of Blood Cells and Blood Cancers, Walter and Eliza Hall Institute of Medical Research, Melbourne, VIC 3052, Australia

**Keywords:** AML, survival, precision medicine, FLT3, IDH, TP53

## Abstract

The better understanding of the genomic landscape in acute myeloid leukaemia (AML) has progressively paved the way for precision medicine in AML. There is a growing number of drugs with novel mechanisms of action and unique side-effect profiles. This review examines the impact of evolving novel therapies on survival in AML and the challenges that ensue.

## 1. Introduction

The last few years have witnessed a therapeutic renaissance in the field of acute myeloid leukemia (AML), spearheaded by the FDA’s approval of 10 new therapies since 2017. While treatment for AML remains a therapeutic challenge, new therapeutic options have begun to create an emerging precision medicine framework in our approach to managing this condition. Along with the growing number of drugs with novel mechanisms of action is the arrival of an array of new adverse event profiles that are often uniquely linked to a particular class of drugs. This review examines the impact of evolving novel therapies on survival in AML and some of the novel toxicities associated with their introduction. We also discuss emerging strategies used to augment and further improve efficacy and the role of measurable residual disease (MRD) as a means for the assessment of clinical outcomes. We propose that the current evidence points to a future of AML management that will be more personalized in terms of treatment selection, safety planning, monitoring for disease response, and the re-evaluation of the mechanisms of adaptive resistance at relapse.

## 2. Targeting Mutated Proteins 

### 2.1. FLT3

FLT3 mutations are one of the most common and prognostically important genetic alterations that occur in AML. FLT3 mutations are present in approximately 30% of all patients with AML, with internal tandem duplication (ITD) accounting for approximately 25% of all AML cases and FLT3 mutation in the tyrosine kinase domain (FLT3-TKD) accounting for approximately 7–10% of all cases. Both mutations lead to the constitutive activation of the receptor and its downstream signalling pathways, including PI3K/AKT/mTOR, RAS/MAPK, and STAT5 (Figure 1). FLT3 mutations cause leukaemic blast survival and proliferation [1]. FLT3-ITD mutations in AML are known unfavorable prognostic markers, as they confer poor overall survival (OS) and relapse-free survival [2].

Targeted therapy with FLT3 inhibitors has significantly improved outcomes in this subgroup. First-generation tyrosine kinase inhibitors (TKIs), such as midostaurin, sorafenib, and lestaurtinib, have demonstrated limited anti-leukemic activity as monotherapies. Midostaurin was shown to improve 4-year OS from 44% to 51% when combined with intensive chemotherapy in patients with untreated FLT3-mutated AML, as demonstrated in a landmark randomized phase 3 study (RATIFY) [4]. Consequently, midostaurin had been approved as first-line therapy for FLT3 mutant AML in combination with intensive induction and consolidation therapy; it has also been approved for use as a single-agent maintenance therapy in some countries. Another first-generation TKI sorafenib, in combination with intensive chemotherapy, has been shown to have anti-leukemic efficacy, with a 2-year event-free survival of 40–60% [5,6]. A recent randomized study reported a relapse-free survival of 61% in the sorafenib group compared to 36% in the placebo group after achieving complete remission [7]. The current standard of care for eligible patients with FLT3-mutant AML is frontline midostaurin in combination with intensive chemotherapy and consideration for allogeneic haematopoietic stem cell transplant (alloHSCT) in first remission, given the high relapse rates after traditional cytotoxic chemotherapy.

New data concerning the role of other FLT3 inhibitors for the first-line treatment of patients fit for intensive chemotherapy are emerging. The Australasian Leukaemia and Lymphoma Group (ALLG) conducted a randomized, placebo-controlled phase 2 trial of sorafenib in combination with intensive induction and consolidation chemotherapy followed by 12 months of sorafenib maintenance. Several themes emerged, including the high rate of clinical response in both the sorafenib and placebo arms (91% vs. 94%); the higher rate of early FLT3-ITD MRD clearance, as assessed by next-generation sequencing (NGS) in the sorafenib arm (43% vs. 32%); the high rate of alloHSCT in both arms (~60%); and the high frequency of FLT3-ITD-negative relapse (~60%) [7]. Although the outcomes were similar to those of prior sorafenib single-arm studies [8], this study did not demonstrate a significant improvement in either EFS or OS for the sorafenib arm. A sub-group analysis showed a trend for improved OS amongst patients with a higher FLT3-ITD allelic ratio (>0.7), which is consistent with the potentially greater oncogenic dependency on FLT3 in this patient population [7]

In the frontline treatment setting, gilteritinib in combination with chemotherapy has also shown favorable anti-leukemic responses, with a composite CR of ~80% and a 70% mutational clearance of FLT3-ITD found in a Phase I study [9]. Based on these results, randomized trials of induction and consolidation chemotherapy plus gilteritinib vs. midostaurin in FLT3-mutant AML patients are underway (NCT04027309) (Table 1). Similar phase III studies of quizartinib vs. placebo (NCT02668653) and crenolanib vs. midostaurin (NCT03258931) in combination with chemotherapy are also ongoing, aiming to provide evidence on the most efficacious FLT3 TKI in newly diagnosed FLT3-mutant AML. In patients unfit for intensive chemotherapy, azacitidine and gilteritinib have yielded promising preliminary results with a 67% CR (n = 15) [10]. Further results of randomized studies are awaited before this combination can be incorporated as standard-of-care practice in the real world. 

Despite the improved OS and EFS found in the RATIFY trial, only 59% of patients in the midostaurin arm achieved CR and almost half of these patients relapsed after achieving remission. AlloHSCT remains beneficial for sustaining long-term remission [11]. A post hoc analysis of RATIFY confirmed that midostaurin conferred survival benefits in FLT3-ITD AML across all three ELN risk groups, irrespective of the FLT3-ITD allelic burden or co-mutation profile (e.g., with or without NPM1 mutation) [11]. The similarly improved outcomes among patients with a low FLT3-ITD allelic burden suggest that some of the clinical benefits associated with this multikinase inhibitor may be mediated through actions targeting other anti-leukemic pathways [11]. The SORAML trial examined the role of combining sorafenib with intensive chemotherapy in patients with AML, not limited to those with the FLT3 mutation. This study demonstrated improved EFS outcomes across a diverse AML cohort, supporting the potential for a biologically relevant off-target activity associated with multikinase inhibitors in combination with chemotherapy in the first line treatment of AML [12].

There is a need for increased awareness of how to anticipate, mitigate, and manage complications associated with the increasing use of FLT3 inhibitors. Severe adverse events associated with the use of multikinase FLT3 inhibitors include midostaurin-related grade 3 to 5 pneumonitis or radiographic pulmonary opacities, which were reported in 8% of patients in the phase RATIFY trial [2]. Among older populations (>60 years) receiving midostaurin in combination with intensive chemotherapy, the risk of cardiac toxicity was reported in 22% of patients, highlighting the importance of ECG and electrolyte monitoring in this patient population. Older patients also experienced more pulmonary adverse events than younger patients did (14% vs. 7%; *p* = 0.07) [13].

In relapses after first-line midostaurin in combination with intensive chemotherapy, different modes of treatment failure have also been noted. A recent study by Schmalbrock et al. provided novel insights into clonal evolution and resistance mechanisms in the RATIFY study. Almost half the midostaurin-treated patients (46%) were *FLT3*-ITD-negative at the time of disease relapse or progression [14]. In other patients, FLT3 selective pressure propagated mutations in the MAPK pathway, potentially bypassing the effect of FLT3 inhibition [14]. 

In relapsed refractory (R/R) AML, second-generation TKIs including quizartinib, crenolanib, and gilteritinib have shown favorable single-agent clinical efficacy, related in part to their enhanced FLT3 target affinity. Gilteritinib was shown to be superior to salvage chemotherapy in the R/R setting by improving the median OS from 5.6 to 9.3 months (*p* = 0.007) in the ADMIRAL study [4]. Gilteritinib is now considered as the treatment of choice for patients with R/R FLT3-mutant AML at first salvage. Adaptive resistance to gilteritinib includes off-target mutations involving the RAS-MAPK pathway and on-target drug resistant FLT3 mutations. In many cases, the adaptive resistance was polyclonal. Furthermore, in approximately half of the relapsing cases, the cause of treatment failure was unknown, indicating that other mechanisms remain to be identified. 

Given the clinical limitations of monotherapy, TKI combination approaches have received considerable research interest. A recent Phase I study found that the combination of gilteritinib and venetoclax in R/R FLT3-mutant AML achieved a modified composite CR (CR + CRp + CRi + MLFS) of 76.4%, including a significant proportion (62.5%) of patients with prior FLT3 TKI exposure, in contrast to 12% in the ADMIRAL study [15]. This suggests the promising anti-leukemic activity of this combination compared to single-agent gilteritinib with the vigilant management of cytopenias using dose interruption and modification [15]. 

Following the attainment of remission, FLT3 TKIs have been actively explored as maintenance therapies for patients with FLT3-mutant AML. However, the impact of maintenance therapy on survival is yet to be ascertained. The beneficial effects of sorafenib maintenance after alloHSCT have been demonstrated based on two randomized trials (NCT02156297, NCT02474290), albeit most patients were not exposed to FLT3 TKIs prior to alloHSCT [16,17]. A systematic review of seven studies (five sorafenib studies and two midostaurin studies) suggested that TKI maintenance therapy after alloHSCT was associated with a marked 65% reduced risk of relapse, as well as improved RFS and OS [18]. These apparent survival and relapse benefits came at the expense of the possible increased risk of acute and chronic graft versus host disease with sorafenib [18]. Prospective randomized trials testing the efficacy of gilteritinib maintenance, following consolidation (NCT02927262), and after alloHSCT (NCT02997202) (Table 1) remain critical to determine the ultimate role of FLT3 TKIs in maintenance therapies and the necessary treatment duration for FLT3-mutant AML. Future studies are required to identify the ideal TKI in terms of efficacy and safety in the post-transplant setting.

### 2.2. IDH1 and IDH2

Isocitrate dehydrogenases (IDHs) are NADP-dependent enzymes involved in cellular energy generation in the Krebs cycle. IDH catalyzes the oxidative decarboxylation of isocitrate to α-ketoglutarate (α-KG). Somatic gain-of function mutations in IDH1 or IDH2 are found in approximately 20% of newly diagnosed AML, with a higher frequency seen in older patients. Mutant forms of IDH1 and IDH2 gain neomorphic function and convert α-KG to oncometabolite 2-hydroxyglutate, which in turn inhibits α-KG-dependent enzymes (Figure 1). This results in pro-leukemic epigenetic changes, including the disruption of TET2 function and the suppression of granulocyte maturation [19,20,21]. 

Ivosidenib and enasidenib, selective inhibitors of mutant IDH1 and IDH2, respectively, have demonstrated encouraging clinical activity for use as monotherapies in the R/R and, more recently, in the frontline setting. In patients with R/R IDH1/2-mutant AML, ivosidenib and enasidenib have induced therapeutic responses in a considerable proportion of patients with an ORR ~40%, CR rates ~20%, and median time to CR of approximately 3 months [22,23]. More recently, Roboz et al. demonstrated that ivosidenib induced deep and durable remissions in treatment-naïve IDH1-mutant AML, with a CR/CRh rate of 42.4%, a mutation clearance in 64%, and in some cases (5/33) a durable remission beyond 2 years [24]. Similar observations have been noted in IDH2-mutant AML using enasidenib with a CR/CRi 30.8% [25]. The median OS achieved with targeted monotherapy in IDH1- and IDH2-mutant untreated AML—12.6 months and 11.3 months, respectively—is favorable compared to that of azacitidine (10.3 months) and decitabine (7.7 months) in a similar patient population [24,25,26,27].

Despite the promising results of early-phase studies, comparative randomized trials with IDH inhibitors have delivered mixed results. The phase III IDHENTIFY trial was planned as the main study to obtain full registration for the IDH2 inhibitor enasidenib. However, this was to no avail, as the study did not demonstrate a superior OS as anticipated. IDHENTIFY was an open-label trial for patients aged ≥60 years who had received 2–3 prior AML therapies. Investigators preselected patients to one of four conventional care regimens (CCR) in the control arm, including either azacitidine, intermediate-dose cytarabine, low-dose cytarabine (LDAC), or best supportive care. Despite the improved ORR found in the Enasidenib arm (40.5% vs. 9.9%), the primary endpoint of OS was not significantly different between the Enasidenib and CCR arms: 6.5 months vs. 6.2 months (HR 0.86; *p*
*=* 0.23) [28]. The study outcome was partially compromised by 12% of patients in the CCR arm receiving post-study enasidenib [28]. Furthermore, the response and potential benefit from enasidenib are likely lower among patients with more advanced disease. In IDH2-mutant AML, the combination of azacitidine and enasidenib vs. azacitidine monotherapy in a phase II study resulted in a significant improvement in response rates (ORR 71% vs. 42%, *p* < 0.01; CR 53% vs. 12%, *p* < 0.01) and durations (median DOR 24.1 months vs. 12.1 months, *p* = 0.05) [29]. Again, the survival outcome was compromised by a 21% post-study crossover in the azacitidine arm to enasidenib [29]. This reflects lessons learned from the design of phase III studies; future studies will therefore aim to focus more on the role of IDH inhibitors at earlier stages of disease and in combination with other drugs. 

Among older patients unfit for intensive chemotherapy, the combination of azacitidine and ivosidenib in a phase Ib study demonstrated an ORR of 78.3%, including CR in 60.9% and mutation clearance in 43.4% [30]. The median duration of response (DoR) in responders was not reached, with median follow-up of 16 months [30]. A phase III AGILE study (NCT03173248) to examine the efficacy of azacytidine and ivosidenib vs. azacitidine only in the frontline setting was stopped early due to the compelling improvement for enhanced outcomes in the ivosidenib arm, including improved EFS, the primary endpoint of the trial, as well as improved overall survival and clinical response [31]. 

Although ivosidenib and enasidenib responses are clinically durable, the complex genetic heterogeneity of AML suggests that combination with other non-targeted therapies may augment remission and overcome primary resistance [32]. Promising preliminary efficacy supports the combination of ivosidenib or enasidenib with intensive chemotherapy (Table 1) in newly diagnosed AML, with composite CR rates (CR + CRi + CRp) of 77% in the ivosidenib group and 74% in the enasidenib group; mutation clearance was also observed in 39% and 23% patients with the IDH1 and IDH2 mutations, respectively [33]. The clearance of FLT3 and RAS mutations in this study supports the role of combining IDH inhibitors with intensive chemotherapy to limit primary resistance and relapses mediated by signaling mutations [33]. 

The emerging role of IDH inhibitors in AML has also led to improved insights into the mechanisms of therapeutic resistance to these agents. An analysis of 179 patients with R/R AML treated with ivosidenib suggested that primary resistance to ivosidenib was related to receptor tyrosine kinase (RTK) pathway mutations such as NRAS and PTPN11 [34]. It was also proposed that acquired resistance at disease relapse after initial response was caused by the outgrowth of off-target clones with the activation of the RTK pathway, as well as the acquisition of 2-HG-restoring mutations, including novel second-site allosteric mutations at the IDH protein–dimer interface either in the mutated IDH gene (mutation in cis) or in the other wild-type IDH1/IDH2 allele (mutation in trans) interrupting the binding efficiency of IDH inhibitor to its cognate binding site [34,35]. Although reductions in the IDH variant allele frequency and molecular clearance were associated with achieving CR, the ability to respond to IDH inhibitors did not correlate with IDH mutation burden [32]. One hypothesis is that targeting mutant IDH cells quenches the effect of the oncometabolite 2-HG from exerting a paracrine effect on neighbouring non-IDH mutant blasts. The biomarkers associated with clinical response remain unclear. In examining the prognostic significance of co-mutations in IDH-mutant AML, the NPM1 mutation was associated with improved OS in IDH1- or IDH2R140-mutated AML treated with intensive chemotherapy, suggesting a potential group of patients who would particularly benefit from the addition of targeted therapy [36].

BCL2 signaling is thought to be a critical survival pathway in IDH1/IDH2-mutant AML based on preclinical studies, providing a rationale for targeted BCL2 inhibition in IDH-mutant AML [37]. Notably, a particularly favorable response was seen in the subgroup of patients with IDH-mutant AML using a combination of BCL2 inhibitor venetoclax and azacitidine (12-month OS 66.8% in venetoclax and azacitidine versus 35.7% in the azacitidine only group) [38]. 

As IDH inhibitors restore myeloid differentiation, they can induce myeloid maturation and proliferation, resulting in IDH inhibitor “differentiation syndrome”. This is reported in ~15% patients and may occur with hyperleucocytosis [24,25]. Often occurring during the initial period of myeloid maturation, it is a non-specific syndrome manifesting as dyspnea, hypoxia, culture-negative fever, pulmonary infiltrates, pleural or pericardial effusions, peripheral edema, and weight gain. A high clinical index of suspicion is required for the timely initiation of corticosteroids, as these symptoms frequently overlap with infections and progressive AML. 

### 2.3. TP53

In AML, the *TP53* mutation accounts for approximately 5–20% of cases. It is associated with a complex and/or monosomal karyotype with very poor survival outcomes after conventional treatment options, with reported CR rates of 20% using hypomethylating agents and a median OS of only 6–12 months [39,40]. *TP53* functions as a tumor suppressor gene and the majority of *TP53* mutations are missense in nature and associated with dominant negative function [41].

Following the recent discovery that *TP53* mutations in myeloid malignancies drive clonal selection through a dominant-negative effect without gain-of-function activities, several novel small molecules to reactivate missense-mutant p53 protein have been investigated to restore the wild-type function of p53 (Figure 1) [41,42,43]. Of these, eprenetapopt (APR-246), a first-in-class mutant p53 reactivator, has received considerable attention due to its early but compelling clinical efficacy when combined with azacitidine in TP53-mutant MDS and AML [43,44,45]. It is a prodrug that is spontaneously converted to active substance methylene quinuclidinone (MQ) under physiological conditions and binds covalently to specific cysteine residues in the mutant p53 to induce apoptosis [42]. 

A recent phase II study (NCT03072043) of azacitidine and eprenatopopt reported an ORR of 73%, with 50% achieving CR in the MDS cohort (n = 40) and an ORR of 64%, with 36% achieving CR in the oligoblastic AML cohort (n = 11) [44]. The median OS was 10.8 months, with significant survival improvement seen in the responders (14.6 vs. 7.5 months) [44]. Comparable rates of clinical response were also demonstrated in another phase II study by Groupe Francophone des Myélodysplasies (GFM) (NCT03588078), with an ORR 63% (47% CR) in MDS and an ORR 33% (17% CR) in oligoblastic AML. Although there were promising phase II results in both AML and MDS, the phase III study in MDS (NCT03745716) did not demonstrate a tbetter CR rate. Phase 3 results in AML are awaited. 

Apart from p53 reactivation, targeting immune evasion is another therapeutic strategy in TP53 mutated AML. CD47 (also known as integrin-associated protein) is a key macrophage immune checkpoint and serves as an anti-phagocytic or “do not eat me” signal [46]. CD47 is overexpressed in myeloid malignancies and its expression is enriched in AML leukaemic stem cells (LSCs) in contrast to their non-malignant counterparts [46,47]. Magrolimab (Hu5F9-G4) is a novel IgG4 antibody that blocks CD47 and induces tumor phagocytosis. Azacitidine synergizes with magrolimab by inducing pro-phagocytic “eat me” signals on leukemic blasts, thus enhancing anti-leukaemic activity [46]. Combination therapy using azacitidine and magrolimab in MDS and AML in Phase Ib study has shown encouraging efficacy. particularly in the TP53-mutant AML, a treatment-refractory group (ORR 64%, CR 41%) [48]. Follow-up results from ongoing expansion cohorts in AML (NCT03248479) are awaited, and a phase III trial evaluating azacitidine and magrolimab in untreated TP53-mutant AML (NCT04778397) is underway to thoroughly evaluate its therapeutic value to augment remission in TP53-mutant AML. 

### 2.4. KMT2A/NPM1

KMT2A gene is located on Ch 11q23.3, with >100 fusion partners identified to cause leukemogenesis. Fusion genes induce aberrant HOXA and MEIS expressions secondary to the recruitment of chromatin-associated complexes, including the enzymes menin and DOT1L, resulting in stem cell-like gene expression signatures [49,50]. Recurrent chromosomal translocations involving the KMT2A gene initiate aggressive forms of leukemias and are often refractory to conventional treatments with a dismal prognosis [49]. AML with NPM1 mutation is also known to share similar stem cell-like gene expression with deregulated HOXA and MEIS genes; therefore, they are also sensitive to chromatin complex inhibition [51]. 

Novel drug discoveries including DOT1L inhibitors, bromodomain inhibitors, and menin inhibitors have demonstrated preclinical efficacy with ongoing validation in prospective clinical studies [50,52,53]. The DOT1L inhibitor pinometostat showed a modest anti-leukaemic activity in patients with KMT2A-rearranged leukemia in a phase I study [53]. More recently MLL-menin binding inhibitors have entered clinical testing as single agents for AML (K0539, NCT04067336 and SNDX-5613, NCT04065399), with great potential to emerge as important targeted therapies in NPM1-mutated and KMT2A-rearranged AML (Figure 1). The preliminary activity of SNDX-5613 in Phase 1 study has been reported with an ORR 48% in heavily pretreated patients harboring an MLL rearrangement or NPM1 mutation [54]. Like other novel agents, the combination of MLL-menin inhibitors with other drugs, including cytotoxic chemotherapy or novel agents, to augment treatment efficacy can potentially be explored. 

### 2.5. Hedgehog/Glioma-Associated Oncogene Homolog (HH-GLI) Signalling Pathway 

The Aberrant Hedgehog (Hh) signalling pathway affects the proliferation of leukaemia stem cells, and its upregulation results in chemo-resistance in AML cell lines [55]. This pathway is regulated by the negative regulator patched (PTCH) and positive regulator smoothened (SMO) [56]. Glasdegib acts as an oral inhibitor of the Hh pathway by interacting with SMO. Its proposed therapeutic potential lies in its suppression of the chemo-resistant stem cell population; hence, clinical relapse can be delayed by inhibiting this pathway. In a randomized phase 2 study of the administration of 20 mg of LDAC subcutaneously twice daily for 10 days with or without glasdegib at 100 mg a day in newly diagnosed AML or high-risk MDS patients, the combination of LDAC and glasdegib was shown to improve OS (8.3 months vs. 4.3 months, HR 0.51; 80% CI, 0.39–0.67, *p* = 0.0004) and CR (15% vs. 2.3%) [57]. This study resulted in the FDA approval of glasdegib when used in combination with LDAC. A small retrospective single-centre review using LDAC and glasdegib in R/R AML, including a 29% relapse of AML post alloHSCT and a 45% venetoclax-treated population, showed a modest CRc of 21% with a median OS of 3.9 months [58]. More robust results from phase 3 randomized studies are awaited to definitively demonstrate the superiority of glasdegib in combination with LDAC (Table 1). The widespread adoption of the LDAC and glasdegib combination in the real world remains uncertain, considering the better response rates demonstrated in other more active combinations—for instance, HMA or LDAC plus venetoclax in the chemotherapy-unfit population. This treatment can be considered in combination with other cytotoxic therapies in the venetoclax-refractory population; however, further data are required. 

## 3. Immunotherapy

Immunotherapy, including alloHSCT and donor lymphocyte infusion, has been used in the treatment of AML for many years, with T cells considered to be the major contributor to the success of this therapy [59]. However, the desirable graft-versus-leukaemia effect of alloHSCT is often offset by the significant and long-term side effects of graft-versus-host disease, restricting the widespread use of alloHSCT to achieve cure in AML. To date, various strategies aiming to harness the robust anti-tumour activity of T cells while limiting T-cell cytotoxicity against healthy tissues have been successful in the treatment of hematological malignancies, mainly B cell disorders. Nevertheless, targeted immunotherapy in AML is challenged by the lack of AML-specific target antigens and the complex clonal architecture, with multiple driver mutations existing in the AML tumour microenvironment [60,61]. 

### 3.1. Antibody-Drug Conjugates (ADCs) 

ADCs consist of three main components: a monoclonal antibody (mAb) that recognizes an antigen target on tumor cells; a cytotoxic molecule, often referred to as a payload, and a chemical linker that connects the mAb and payload [62]. They are designed to bind selectively to antigen-positive cells, enabling the targeted delivery of cytotoxic payload to tumor cells. 

CD33 is a transmembrane receptor expressed by committed myeloid cells, including the majority of leukemic blasts but not normal hematopoietic stem cells [63]. Hence, CD33 is an attractive target in the treatment of AML. Gemtuzumab ozogamicin (GO) is a humanized anti-CD33 conjugated to a calicheamicin derivative; once released intracellularly, it leads to DNA binding and cell death [63]. The combination of GO and chemotherapy has been shown to improve RFS and OS, especially in patients with favorable-risk core-binding factor AML and intermediate-risk disease, including NPM1-mutated AML, but not in patients with adverse-risk disease, likely related to the high levels of CD33 expression [64,65,66]. The addition of GO to intensive chemotherapy in NPM1-mutated AML has been shown to lead to better clearance of the NPM1 transcript level, resulting in a lower relapse rate (4-yr CIR 29.3% vs. 45.7%, *p* = 0.009) [64]. Overall, the prolongation of EFS is attributed to the depth of response and the prolongation of CR with the addition of GO [66]. The addition of GO represents a worthwhile option in the frontline setting for CD33-expressing disease [66,67]. However, the use of GO requires prompt recognition and management due to the increased risk of hepatic veno-occlusive disease/sinusoidal obstruction syndrome (VOD/SOS), especially following hematopoietic stem cell transplantation, prolonged thrombocytopenia, infusion-related reaction, and tumor lysis syndrome. 

### 3.2. Bispecific Antibodies

Bispecific antibodies are recombinant protein constructs that engage T cells through CD3 and bind to tumor-associated antigen, typically with a higher affinity. In AML, potential therapeutic targets are lineage-restricted antigens including CD33, CD123, CLL-1, and FLT3, for which early phase clinical trials are underway. 

CD123 (also known as interleukin 3 receptor α-chain, or IL3RA) is expressed on normal haematopoietic stem/progenitor cells, but it is expressed more on AML blasts and LSCs [68]. It is an attractive therapeutic target in AML, as high CD123 expression on blasts confers poor outcomes and high CD123 expression is also enriched in primary induction failure/early-relapse AML [68,69,70]. The most advanced CD123-based immunotherapy currently in clinical development is flotetuzumab, a CD123 x CD3 dual-affinity retargeting (DART) antibody. Flotetuzumab has shown preliminary efficacy in R/R AML (NCT02152956), with a CR/CRh of 20% and a satisfactory safety profile [71]. It is of note that clinical activity was mainly observed in patients with primary induction failure (PIF) or early relapse <6 months (ER), a chemo-refractory group showing a CR/CRh of 27% and an ORR of 30%, with a median OS of 10.2 months and 6- and 12-month survival rates of 75% and 50%, respectively [71]. This is encouraging compared to the dismal outcome with an ORR ~12% for salvage chemotherapy in patients with PIF/ER and a median OS of 3 months [72,73]. 

The observed response of patients with PIF/ER AML to CD123-targeted immunotherapy is in accordance with the finding that an immune-infiltrated (IFN-γ–dominant) tumour microenvironment can identify patients who are less likely to respond to cytotoxic chemotherapy but more likely to respond to immunotherapy [74]. In comparison to B-cell immunotherapies, the incidence and severity of infusion-related reaction (IRR)/cytokine release syndrome (CRS) using CD123-targeted immunotherapy may be more pronounced (96% IRR/CRS but only 8% Grade 3–4) due to the shared target antigen expression of monocytes and macrophages, which mediate IL-6 production [71]. Stepwise dosing, pre-medication with dexamethasone, the prompt use of tocilizumab, and temporary dose reductions or interruptions have been shown to prevent severe IRR/CRS [71]. In summary, immunotherapy targeting CD123/CD3 may offer a novel treatment option in PIF/ER AML. The expression of CD123 by leukemic stem cells further strengthens the potential therapeutic value of this target [70,75]. 

Several potential biomarkers used to predict responders in CD123-targeted immunotherapy have been identified, including a higher CD123 receptor density and higher CD123 mRNA using gene expression profiling in patients with PIF/ER [71]. Future studies on measurable residual disease and/or leukemic stem cell eradication, as well as its impact on survival outcome using immunotherapy, are warranted in order to allow its incorporation as a frontline treatment.

T-cell-directed therapy using bispecific antibodies is a promising immunologic approach in the treatment of AML. However, the most suitable and appropriate epitopes for limiting on-target off-tumor toxicity in AML are yet to be identified. One of the proposed ways to optimize the success of bispecific antibodies in AML is to test it earlier in the treatment sequence—for instance, in the first salvage or in the MRD setting—to minimize T-cell exhaustion, as preserved T-cell function is critical for the activity of bispecific antibodies [61]. To improve survival outcomes and ensure the safe delivery of immunotherapy, ongoing efforts to determine biomarkers that will help identify the patients most likely to benefit from immunotherapy, to determine the ideal timing of therapy (e.g., frontline, MRD positive, maintenance, early salvage), and to determine the optimal combination partners and/or sequence are vital [61].

### 3.3. Chimeric Antigen Receptor (CAR) T Cell Therapy

Unlike the successes of CAR-T cell therapy in B cell malignancies, the progress of the use of CAR-T cells in the treatment of AML has been hindered, mainly due to the lack of suitable targetable antigens and the poorly tolerated consequences of complete myeloid progeny ablation [76,77]. Most of the AML antigens targeted by CAR-T cells are frequently expressed in normal hematopoietic stem/progenitor cells and other organs, including the lungs and liver, causing an increased risk of on-target off-tumor toxicity [61]. Early-phase AML CAR-T and CAR NK clinical trials targeting CD33, CD123, and NKG2D are ongoing, with creative solutions that actively seek to overcome therapeutic obstacles [61]. The impact of CAR-T therapy in AML is awaited with cautious optimism.

## 4. Conclusions

The powerful diagnostic technologies created in recent years as part of the therapeutic revolution have transformed the paradigm of AML treatment, with increasing focus now placed on precision medicine approaches. Targeting the disease heterogeneity of AML has provided a more rational therapeutic approach compared to the ‘one-size-fits-all’ approach using conventional chemotherapy. It is likely that, in the long term, targeted agents will be used in combination with chemotherapy to eliminate measurable residual disease and for maintenance treatment. The potential areas of focus for future research in AML therapy are summarized in Table 2, with specific areas for each molecular aberrations highlighted and discussed throughout the paper. The ongoing challenges will be to predict individualized treatment responses using biomarkers as well as to translate the initial responses into deep and durable disease control by incorporating these novel agents into upfront treatment.

## Figures and Tables

**Figure 1 jpm-11-01003-f001:**
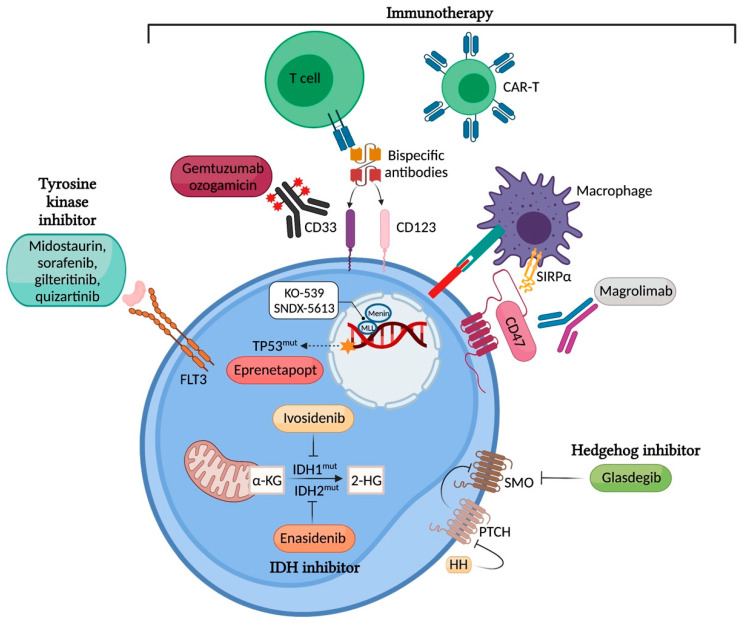
Targeting mutated proteins in AML. Image created with BioRender.com [3] (accessed on 26 September 2021).

**Table 1 jpm-11-01003-t001:** Randomized clinical trials for drug development in AML.

Class of Drugs	Investigated Agent	Investigation	Trial Registration Number
FLT3 inhibitors	Gilteritinib	Phase III, frontline, gilteritinib vs. midostaurin in combination with “7 + 3”, FLT3 mut AML	NCT04027309
		Phase III, maintenance, gilteritinib vs. placebo, FLT3 mut AML in CR1 after chemotherapy	NCT02927262
		Phase III, maintenance, gilteritinib vs. placebo, FLT3 mut AML after allogeneic haematopoietic stem cell transplantation	NCT02997202
	Quizartinib	Phase III, frontline, quizartinib vs. placebo in combination with “7 + 3”, FLT3 mut AML	NCT02668653
	Crenolanib	Phase III, frontline, crenolanib vs. midostaurin in combination with “7 + 3”, FLT3 mut AML	NCT03258931
IDH1 inhibitors	Ivosidenib	Phase III, frontline, ivosidenib vs. placebo in combination with “7 + 3”, IDH1 mut AML	NCT03839771
		Phase III, frontline, ivosidenib vs. placebo in combination with azacitidine, IDH1 mut AML	NCT03173248
IDH2 inhibitor	Enasidenib	Phase III, frontline, enasidenib vs. placebo in combination with “7 + 3”, IDH2 mut AML	NCT03839771
		Phase III, enasidenib vs. conventional care, ≥60 years, late stage IDH2 mut AML	NCT02577406
Hedgehog inhibitor	Glasdegib	Phase III, frontline, glasdegib vs. placebo in combination with “7 + 3” or azacitidine, fit and unfit patients	NCT03416179

Abbreviation: “7 + 3”, intensive induction chemotherapy with 7 days of cytarabine and 3 days of anthracycline.

**Table 2 jpm-11-01003-t002:** Potential areas for future research.

Target/Therapy	Areas of Focus for Future and Ongoing Research
FLT3	Role of 2^nd^-generation inhibitors in the front-line setting. Identifying an effective combination with other novel agents. Role of maintenance therapy post chemotherapy and post allogeneic transplant. Potential for targeted therapies to eliminate MRD. Abrogating clonal evolution and resistance following frontline TKIs.
IDH1/2	Role of IDH inhibitors in combination with frontline intensive chemotherapy. Identifying optimal combinations with novel agents. Describing the biomarkers of response and survival—does mutation clearance have an impact? Understanding the mechanisms of primary and acquired resistance. Establishing the role of IDH inhibitors in the maintenance setting.
p53 and KMT2A	Identifying effective targeted agent and combinations.
Immunotherapy	Single versus combination therapy. Defining the role in the initial treatment of AML. Establishing the role in maintenance and MRD elimination.

## Data Availability

Not applicable.

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
