# Peer review of "New Drugs Bringing New Challenges to AML: A Brief Review"

_jpm, 2021, doi:10.3390/jpm11101003_

Round 1

Reviewer 1 Report

The review is generally well written and illustrates the key therapeutic advances in the management of AML. However, there seems to be a lack of focus in that many therapeutic options are touched but without in-depth coverage and clear definition of terms. The authors may consider expand their key concepts and categorize the treatment options based on different treatment settings (upfront induction vs. therapy for reinduction vs. treatment for relapsed or refractory disease).  A clinically relevant discussion on the real world practice is needed, rather than a summary of all clinical trials. 

Author Response

Thank you for your constructive feedback. Key concepts had been expanded and paper had been restructured to present treatment options in the sequence of frontline, relapsed refractory and maintenance setting, in a more explicit way. We have highlighted key concepts and synthesised clinical trial findings where possible. Whilst there are many emerging therapeutic options in AML, many of these are early phase studies with more mature clinical trials underway. This does limit our ability to provide practice-changing clinical recommendation at this stage. Hope this explains the way this review paper has been written. 

Reviewer 2 Report

In this Review entitled “New drugs bringing new challenges to AML: a brief review”, Yeoh and colleagues discussed 10 new therapies approved by FDA for AML since 2017, focusing on the available clinical trial results for each therapy. This Review has both scientific value and clinical relevance.

The reviewer has the following suggestions for the authors’ consideration:

  1. Providing a figure to summarize the key information of this review article will help the readers to quickly locate the information of interest in the manuscript;
  2. The potential areas of future research listed in Table 1 need to be discussed in the “conclusion” part of the article.
  3. Please make sure Table 2 is cited in the manuscript.

Author Response

Thank you for your constructive feedback. 

  1. Figure provided. 
  2. The potential areas of future research listed in Table 1 had been discussed throughout the paper in the relevant sections. 
  3. Table 2 has been cited in the manuscript.